# Analysis of Religious Socialization Based on Interviews Conducted with Young Adults

**Gabriella Pusztai** and **Zsuzsanna Demeter-Karászi** *

Institute of Educational Studies and Cultural Management, University of Debrecen, 4032 Debrecen, Hungary;
gabriela.pusztai@gmail.com
* Correspondence: zsuzsanna.karaszi@gmail.com

**Abstract:** The term religious socialization has become a pressing issue in the context of religious socialization research. It also raises the question whether religious transmission can be interpreted through the reproduction or constructivist approach. Previously, the reconstruction models determined the approach of studying religious socialization. According to these models, socialization meant the adoption of the patterns of religious practice in the family. In this sense, socialization is periodical. The constructivist and the social network models of sociology significantly changed our conception of religious socialization. The earlier model was replaced by a model which rests on activity, correlation and open-endedness. In this paper, 18 qualitative interviews were analyzeanalyzed. Because religiosity is a multidimensional phenomenon, we wanted to analyze development in each dimension, which is why we relied on Glock and Stark's model. Based on our results, seven types emerged and these findings have strengthened the constructivist approach.

**Keywords:** religious socialization; religious transmission; religiosity

## 1. Introduction

Previously the one-way linear reconstruction models determined the approach of studying religious socialization in Hungarian research on religiosity (Pusztai 2006; Kiss 2007; Hámori and Rosta 2013). The reason for this is that the institutional forms of religious education were limited in the decades of communism in the Central European region. Contrary to expectations, the end of these restrictions on institutional religious education after democratic transformation did not lead to fully effective religious transmission as a result. According to the structural-functional approach, socialization entails following the ready-made social structures of the family and religious practice, while integrating into the laid down institutional framework, making socialization a one-way linear influence. In this perspective socialization is periodical, focusing on one stage of life, with the outcome that the new generation integrates into the traditional religious pattern, and their behavior corresponds to that expected from the people belonging to that particular religious community. According to the critical approach, socialization agents act with power and compulsion to reconstruct religion in a new generation, which can lead to denial and rebellion (Martin et al. 2003). In recent years, the constructivist and social network models of socialization have significantly reshaped our understanding of religious socialization. The linear, teleological and passive model was replaced by the active, correlational and open-ended model, which depends on individual interpretation (Sherkat 2007). Religious socialization, bearing features of group social learning, is characterized by the construction of religious, spiritual content and experiences, as well as by individual and social construction of religious practice. In the earlier model the basic, significant interrelatedness between the patterns of religious socialization, demographic features and socioeconomic status were considered to be dominant, while the constructivist religious socialization model emphasizes local communities

surrounding the individual, the diversity of life goals and styles of life and the fragmentation of the lifecycle, as well as changes as permanent processes.

The current research analyzes 18 qualitative interviews with young adults. We were trying to find out whether earlier religious socialization theories can provide a framework for the characterization of the current young generation and the interpretation of religious changes, or whether it is the active, creative, reconstructing religiosity that better describes the phenomena under scrutiny. Since religiosity is a multidimensional phenomenon, and we wished to investigate development in each dimension, we based our analyses on Glock and Stark's (1965) model. As a result, based on the interviewees' religious socialization we distinguished seven types of religiosity.

## 2. Religious Transmission

Religious socialization is an interactive process within which various agents of socialization influence the development of the individual's religious convictions. Throughout their life, individuals encounter a number of different socialization platforms which influence the development of their religiosity (Sherkat 2007). According to de Hart (1990) nothing can surpass the importance of the family environment in the individual's development of religious socialization. He enumerates the factors influencing religiosity in the following order: the first factor is religious practice within the family, the second is formal religious education, and the third is the denominational school (Pusztai 2006).

There is controversy over the influence of family religiosity, and the successful and less successful transmission of religiosity. However, scholars mostly agree that the family plays a critical role in the representation of religious values and practices (Hayes and Pittelkow 1993).

There is on-going debate on whether the mother or the father is more influential, while others think that the parents play different roles in the different dimensions of religious socialization (Hoge and Petrillo 1979; Hayes and Pittelkow 1993). Through further research, we found that since mothers are religious to a higher rate, the successful transmission of religiosity does not depend on the religiosity of the mother, but rather it is the religiosity of the father that becomes the differentiating factor. The homogamy of the parents' value system and religious worldview is a significant, independent influencing factor. Agreement between parents leads to a significantly more efficient religious transmission. In the contrary case, the religious commitment of the offspring is weaker and when there is religious heterogamy, the father has a stronger influence (Hoge and Petrillo 1979).

Based on the literature of the field, it can be stated that the family plays a significant role in the transmission of religious values, and the success of religious socialization depends partly, but not entirely, on how unified the parents' religion, religiosity, or system of beliefs is.

## 3. Approaches to Religious Socialization

Besides the family as the foremost socialization factor, the literature in the field defines other important agents. One of these agents is church communities; the other is the contemporary social network (Martin et al. 2003). Scholars mostly agree upon the importance of these agents. The composition of the social network becomes of crucial importance in the development of religious views, especially in the case of teenagers (Pusztai 2006, 2015). According to some researchers, the contemporary or institutional influence is in fact the transmitter of the influence of the parent's religiosity (Desrosiers et al. 2011; Martin et al. 2003).

Stages of life also have a role to play. James W. Fowler distinguishes six plus one stages in the development of faith. According to him, faith follows a universal and continuous development across the individual's lifecycle (Fowler and Loder 1982). Building on Erikson's theory, he emphasizes the life-long development of religiosity, pointing out that different stages and crisis of life have an influence on the individual's religious socialization. One of the most important stages of life is young adulthood (Ozorak 1989). This is the stage of life when the individual encounters new media of socialization within the school system, higher education or workplace. Previous research has confirmed the hypothesis

that influences within the institution of higher education play a significant role in the individual's socialization (Pusztai 2015).

For a long time, the individual's religious socialization and the universal theory of socialization were both defined by the reconstruction model (Pusztai 2015), the cornerstone of which is the transmission of ready-made patterns. In this perspective, the salient agents of socialization are the parents, the church and the school. Individuals embrace behavior patterns seen and learned from others and following, in particular, the older generation, they passively take in a permanently sealed "curriculum". With the refinement of socialization models, the reconstruction model was replaced by the constructivist approach, which was also applied to the interpretation of religious socialization. The insecurity caused by the diversity of the religious content forces a continuous reconstruction, and the individuals reconstruct and refine their religiosity by applying self-criticism to their choices regarding their own religiosity (Martí 2015; Hunt 2015). In the constructivist model, the individuals have a creative role. This means they do not take over a ready-made pattern, but instead criticize, select and innovate. The salient agents of socialization are either chosen by the individuals or by the members of their community, and the earlier developed religiosity can be considered from a completely new perspective. The constructivist model can further be classified into two subtypes. In one subtype, the individuals repeatedly reconstruct their religious identity in the given situation, while in the other, religious identity is a social construction, and depends on the individual's community membership. Both subtypes are relevant in the case of the young generation, based on the fact that the other students, and also institutional effects could change or could influence the religious identity of the individuals.

## 4. The Source of the Multidimensional Analysis

According to the multidimensional approach, religious commitment cannot be defined only by various frequency indicators, such as the frequency of churchgoing, or praying (Földvári and Rosta 1998). Glock and Stark's five—dimensional religion model is not recent, however, it provides a great structure for the exceptionally complex phenomenon of religion, dividing it into individual areas identified as the dimensions of religiosity. Separating the ideological, the ritualistic, the experiential, the intellectual, as well as, the consequential dimension confirmed the validity of the analysis, and it is also suitable for comparing religious communities (Földvári and Rosta 1998; Pusztai 2006). The ideological dimension concerns faith and religious beliefs turned into personal convictions. The ritualistic dimension refers to religious practice, a religious activity mostly performed communally. The third dimension comprises the individual's religious experiences. The intellectual dimension measures religious knowledge, while the consequential dimension investigates the influence of the previous four dimensions on the behavioral patterns of the individual. These dimensions can further be broken down into sub-dimensions. One of them is the individual's expectations from religion; the other is how religiosity surfaces in the performance of the individual. It is also important to highlight that the dimensions are independent of one another, if an individual proves to be religious based on a dimension, it does not mean that the individual has to be religious according to another dimension. On the whole, the model is well-suited for illustrating the complexity of religiosity (Földvári and Rosta 1998; Pusztai 2006).

## 5. Research

Further research such as the Hungarian Youth 2012 or the European Social Survey 2012 demonstrate that a growing number of questions remain unanswered about religiosity (Földvári 2014). Several possible reasons for this could be mentioned, for example the individuals may feel that these kinds of questions are private matters, or it is also could be that for such questions there is no appropriate answer (Földvári 2014). During our research we were faced with this problem, namely a significant majority of the young generation rejected invitation to be interviewed and they also added that regarding this topic they refuse to comment.

The database of our research is based on a corpus of 48 in-depth interviews carried out by the members of a research group. The interviewees were selected by using the snowball sampling research

method. As a result of this method, the interviewees included individuals from all regions of Hungary. In case of selection, the research team took into account that the individuals should belong to different groups, in terms of how determinant their religious affiliation is (the individuals are religious according to church teachings, they are religious in their own way or they are not religious at all). The interviews were based on a semi-structured interview scheme and the individuals made use the free possibility to respond in an open manner to questions. The semi-structured interview is also the result of joint editing. The interviews were articulated around five major themes, which all explored themes of religiosity. The first questions were on religiosity during childhood and the questions were worded as follows: How would you describe your parents' religiosity? How often did you go to church? etc. The second set of questions were on religiosity during primary and secondary school and contained these questions: Have you talked about religious topics with your classmates or with your teachers? etc. The third set of questions were on religiosity in young adulthood and consisted of the following questions: Have your religiosity changed when you moved away from your parents? Have you talked about religious topics with your colleagues at work or at university? etc. The fourth set of questions were on individual religiosity and contained questions such as: Do you think you are religious? How do you express your religiosity? etc. And last, but not least the fifth set of questions were on the future plans with questions such as: Did you have a church wedding? Would you like to have a church wedding? Did you baptize your child/children? Would you like to baptize your child/children? If yes, why? If no, why? etc. The interviews investigated three generations of the family in order to study inter- and intragenerational religious changes. The present research focuses on 18 interviews, with 10 females and 8 males, focusing on the youngest generation. The individuals, aged 18–30, belong to the Catholic and Reformed church. Regarding their family status, almost everyone had a relationship (married or registered partnership) and just two individuals were single. In terms of their educational level was also diverse. Ten individuals were students on different faculties, five individuals had university degrees and three individuals had obtained a baccalaureate qualification. The interview questions made it possible to explore the development of a religious lifecycle through the different stages of life. When investigating the religiosity of the young generation by analyzing the interviews, we wanted to separate the phenomena belonging to different dimensions of religiosity because we noticed that change and stability can be present simultaneously in different dimensions. One of the results of the analysis is that through the observation of this extremely complex process, we created types for the interview subjects by using content analysis, by using an open coding method developed by the authors, by reading the transcripts and by coding the material. We distinguished between the following seven types: conveniently keeping (2 interviews), building autonomous religiosity (3 interviews), religiously impaired (2 interviews), drawing near (3 interviews), unreflective nostalgic (2 interviews), uninterested (3 interviews) and self-consciously non-religious (3 interviews).

We created these types in order to answer the following research questions: Our first research question was whether the phenomenon of losing religion can be a new form of religious socialization, which makes the individuals formulate their relation to religiosity in a more critical, independent and creative way. Our second question was whether the reconstructing religiosity socialization process surfaces in the course of the young generation's religious socialization. We analyzed each one of Glock and Stark's dimensions individually in order to determine the dimensions characterized by the inter- and intragenerational stability and the ones characterized by innovation.

## 6. Results

While carrying out the research, we investigated the literature in the field and compared and contrasted it with the types we created. In doing this it became evident that the religiosity of the young generation cannot be constrained by a narrow framework, since we are facing an active, innovative, reconstructing religiosity, with varying degrees of changes with each individual.

The interview subjects belonging to the conveniently keeping type were raised in an actively religious environment. *"My mother and grandmother taught me to pray; we have attended church almost*

*every Sunday since I was a child"* (female). Since their family has been strongly embedded in the local religious social network, they feel at ease in it and this sense of belonging also defines their free time activities and their relationships: *"I eagerly took part in them (camps, religion class, etc.). I liked the people, the tasks, and the atmosphere. Everything was great, we learned through play. I'm a Catholic but the Reformed church also organized such camps and we were welcome there, as well"* (female). Between childhood and adulthood they reflect on their inherited religiosity: they accept the religiosity of the older generation (parents, grandparents), however, they treat it with nuance and tailor it.

Their religiosity is characterized by thorough knowledge and a theoretical background due to religious education within and outside the family. Their worldview is similar to the worldview of the previous generation, however, they emphasize the fact that they cannot follow their parents' discipline and adherence to constraints. Since they take on ready-made religious knowledge, traditions and norms, they are born or grow into religiosity; the conversion experience is very scarce. They identify fully emotional events as religious experiences; however, these were not instances of change, but rather confirmation of the inherited religious system. On the whole, their experiences are positive. *"If you go to church, you'll feel better"* (female).

While they speak about ideological stability, the need for change surfaces in two dimensions. On one hand, and we consider this one to be more important, they would like to relax religious norms: *" . . . these do not really evolve in time. For example, the Catholic church does not accept divorce. I think this is a bit old fashioned. Or living together with somebody before marriage, this is not the worst thing in the world, especially if you cannot divorce. Then you should marry someone you can live with"* (female).

*"A bit different . . . Of course, there are many things I acknowledge in hindsight, I think they were important. True. But there are many things I would change about religion. And I think that is normal, to some extent"* (male). Their needs differ from that of their parents also in the ritualistic dimension. *"I don't attend each mass. I go when I can and I don't feel bad about this"* (female). It needs to be emphasized that the interview material on the parent's lifecycle shows that at the time of the interview the strongly committed and ritualistically very active adults also reported periods of different levels of activity. Thus, we consider the slight change related to norms to be a more important characteristic. The conveniently keeping type is characterized by keeping the ideological and intellectual dimension, the experiential dimension is dominated by stability, the ritualistic dimension is made more convenient, while in some aspects of the consequential dimension, the inherited patterns are relaxed.

The building autonomous religiosity type isn't too different from the previous type. These individuals were also raised under a systematic religiosity which defined their way of thinking and the traditions of the family. They also received a formal religious education. The basic commitment in the ideological dimension is stable. Due to their rich and varied religious education within and outside the family, they have gained diverse and also practical knowledge. They keep the religiosity inherited from the older generation, however, they strongly reflect on nearly every element of the inherited culture. They strongly rationalize religious knowledge, consequently, they redefine religiosity in a functional perspective: *"For a long time it doesn't even cross your mind to doubt it. Evidently, what your teacher says is of course, true. Yes. And when you reach an age at which you question these, perhaps at that time it was less important. These details become less important in the sense that it is the symbolic nature which is more important. So, it has become less important whether he was actually raised from the death, or what happened two thousand years ago"* (male).

The development of the reflective religiosity in this type requires an environment which allowed questions, doubts, criticism, and these became debatable alternatives in the transfer period. In one of the cases, this was connected to the experiences gained within the framework of formal religious education, and the excerpt below illustrates the consolidating role of debates. *"The most open discussion took place towards the end of our secondary-school education, when our class was assigned a priest who treated us like adults, and he was the one who was asking questions. This was rather a discussion than just a series of questions. And we stated our opinions relatively openly"* (male). With this type a certain intellectual quest emerges by the end of post-adolescence, which encompasses religious perspective as a value: *"To know*

*that there is a peaceful, objective view of the future, grounded in the fact that we know there is someone who takes care of us. This is a valuable thought"* (male).

At the same time, a criticism emerges in the ideological and ritualistic dimensions of the elements adopted through early religious socialization: *"There are some key aspects in which I cannot accept the standpoint of the Catholic church. For example, I don't find religious formalities important, such as ornate chasubles or communion. I don't agree with the standpoint that we are born sinners. I think this could be thought over, could be reviewed on a certain level"* (male). In ritualistic questions the need that surfaces is not only one of making things more convenient, shaping them to fit the dominant style of life, as with the previous type, but recognizing the ambiguous aspect of rites, and re-ranking the meanings as compared to the ones adopted during early socialization: *"This (i.e., communion) is rather a partaking in a joint activity than appropriating the sacraments. This is what I find positive in it."* For this type, the expectations regarding religious consequences remind us of traditional patterns, but with a novel, reflected definition: *"It doesn't make me more self-critical, or more self-confident in the traditional sense, that I have more self-esteem. I would rather say that it makes me more optimistic"*. We cannot overlook the particularity that all interview subjects of this type were male.

Individuals belonging to the religiously impaired type have also received religious education, i.e., the family socialization is very similar to the previous two types. The parents' generation might be even more embedded in the local religious social network and they are more active within the religious community. The parents belong to the active core of the community and spend their free time within the community: *"there are some important holidays also during the week and they attend those, as well. They always take part in church life"* (male). *"We have always attended church trips, family days, and the like"* (female). These youngsters have been part of an intensive community life from their early childhood, however, within the family, religiosity was less manifested; there is no sign of interpretation, reflection or questions. One of the interview subjects said that it was the ritualized, inadequate religious practice at a denominational secondary school that intensified her alienation: *"I never wanted to attend a religious school but this is how it turned out. Basically, we are Reformed, and I attended a Catholic school for 4 years. Since the traditions are different, I wasn't willing to adhere to the Catholic traditions"* (female). The youngsters between childhood and adulthood assess religious transmission as a failure: *"Yes, as if I have taken a 180 degree turn. I am totally against religion because they forced it on me"* (female). The most dramatic failure surfaces in the ritualistic dimension: *"What mainly triggered it was neglect. I was not interested, I don't want to go, I don't want to do this, I prefer to be somewhere else . . . it was boring. I didn't want to be there at all. I considered it a waste of time"* (male). *"Sometimes it's agony, sometimes it's pleasure ( . . . ) I won't force religion on my children, I wouldn't drag them to church every Sunday. If they want to come, I will take them. But if it's only torture for them, I won't force it because that would turn them away for a lifetime"* (female).

Religiosity in adolescence is also marked by a rebellion against parents. On one hand they break away from the ideal, unquestionable, so called perfect religiosity of the parents, which becomes a sharp opposition primarily in the ritualistic dimension. *"And now they attend church and I don't. They pray, I don't. But this does not generate any discrepancies between us. And it didn't affect our relationship. I am not against Christianity as a religion. The fact that I am non-religious doesn't mean that I am against it. I just look at things from another perspective. That's all"* (male). Individuals belonging to this type try to step out of the family religious socialization framework, looking for an individual course—*"I am partly religious"* (female)—, and this quest becomes permanent: *"Yes, I believe in a God but I don't know if this is that God. I believe that there is a spirit above us, encompassing time and space—so I would say, if I call this God, then this is almost the same to what Christianity says, that God is here in every particle of this time-space continuum, because he is it, and you are it, because he contains you and vice-versa. This is what I believe in, but I don't necessarily accept the stories presented in the New Testament"* (male). Ideological innovation in the restructuring of the content of faith is not entirely consistent. Interviewees belonging to this type cannot construct new structures: *"I don't really know what I thought. At that time, I probably didn't think anything, I just felt bored and I didn't want to be there. After that, when I became more objective, more detached,*

*I realized that I had made a good choice because I had to experience it, to distance myself from it and have a look at it from a distance"* (male).

Among religious consequences, which in spite of the distancing are positively formulated, some values mentioned are more general, others are more vague. We consider that the banality of these bear the marks of unreflectedness, of a critical stage not dealt with: *"What I like in religion is this eternal striving for peace, I still find this important. I know this is banality but equality and righteousness is what I still find important within a community"* (male). Another interviewee gives a less certain answer*: "For example, somebody who is religious can be a better person, but those who are not religious could also be good"* (female).

Interview subjects belonging to the drawing near type have a radically different motivation. They lacked religious education in their childhood and drew nearer to religiosity in the process of becoming independent adults. They also lacked religious knowledge. *"I first heard about Jesus when I was twelve. I heard about who he was but nothing else"* (male). The unexpected, defining experiences within the religious social network have the most dramatic role with this type. The experience triggers a change in their worldview, it triggers involvement, bonding, embedding, learning, and is often followed by leadership and innovation. These individuals who lacked religion in childhood experience a complete identity transformation in adolescence and post-adolescence. The conversion experience always happens as a community experience: *"It went on by being invited to that community. There was a religion teacher who taught every Friday and invited me to the classes. Basically, it was good. I first went there later, when I was 16 or 17. It was great. We went there on Friday and they prayed, we prayed, read from the Bible, discussed things"* (male). *"I mainly attended religion classes because it was advertised and a great community was formed. I always had a great time, it filled me up, made me happy"* (female). The binding force of the community becomes a stable value, thus the ritualistic dimension in not criticized; on the contrary, it becomes ground for meaningful experiences. With a significant turn from complete lack of religious knowledge and culture, these individuals commit themselves to looking for religious experiences and learning: *"After some time my relationship with God changed, I became more involved in religion than before . . . I read the Bible more, prayed more, I took religion seriously"* (female). Their religious practice culminates with becoming involved in the community. Going through a number of developmental stages, eventually the individual becomes an active organizer: *"We came up with the idea to build a community, to start something, so we formed this group and we get together on Mondays. We meet at six and from six to eight we pray, discuss, sometimes we invite speakers or a priest"* (male). As they proceed through religious change, they do not only look for new friends, but also for the guidance of a religion teacher or priest/preacher and they meet up with professional spiritual pastoral cares and community organizers. The drawing near type is characterized by a dynamic change in each dimension. Conversion has an impact on their social network, life goals, and even their job, while they are not exempted from conflicts arising from the change.

The religious socialization of the individuals belonging to the unreflective nostalgic type is mostly characterized by tangential impact, superficial attraction and sporadic interest. They do have some knowledge related to religiosity, which they generally acquired within the family, from their grandparents. They are also uncertain about the facts relating to their religious lifecycle (e.g., they do not know if they have been baptized or not). They started school following the regime change and consequently their experience of religious education was rather short and positive. However, the acquired knowledge is not systematic, and it cannot even be deployed as cultural knowledge. Their relationship towards religiosity is characterized by a kind of lingering want of something, a religious need arising from time to time, however this remains unreflected. This phenomenon is echoed by the following question posed by one of the interviewees *"There has to be providence, doesn't it?"* (female). The desire for religiosity surfaces in a hesitating ideological orientation: *"Things that happen, happen for a reason, don't they? They are written. Our path in life, the reason we are here, the reason we are born for"* (female). The ritualistic dimension of their religiosity has not developed, they do not have a denominational identity and they are not associated with churches or communities. The strong impact of the social network was mostly palpable with this type since their religious practice depends on the



constitution of their social network. Besides the social network, it was critical life situations that had an impact on their religious practice. *"Yes, because at first it wasn't successful, we were trying. Not for a long time, but not this way, and then noting, and after that there were times when I prayed for help"* (female). Their religious experiences are also connected to answering prayers. They synthetize their expectations of religious consequences mainly in the regulation of behavioral norms. *"Education, the known education system is actually equivalent to a certain extent, because, if you go, you will also receive education"* (female).

Those belonging to the uninterested type are indifferent not only towards religiosity, but also towards the basic questions of existence. Even if these questions arise, they discuss them with a kind of playful, hedonistic attitude. Their parents also lacked religious education, consequently these questions were not raised in the family, and the individuals were not impacted by religious socialization from other sources. From this point of view they are characterized by stability. When it comes to basic questions of existence they operate with not too sophisticated, rather naïve alternative conceptual models: *"I always believed in luck, the question is whether I am lucky or not at that particular moment"* (male).

They differ from the unreflective nostalgic type in their disinterest: *"I don't find it important; I am simply not interested in it. Things will always turn out somehow, how they turn out, we will see"* (male). Individuals belonging to the uninterested type lack systematic religious knowledge. The desire for spirituality is satisfied with a kind of consumable spirituality, such as mediums or horoscopes. When asked whether they believe in some kind of transcendental phenomenon or principle of organization, they all gave negative answers, however, the detailed answers show that they are making an effort to conceptualize these. *"Not necessarily, some of the things were true but it was just a coincidence. I find it strange that they say that someone can always know things; however, this might be due to the fact that a zodiac sign is assigned to a certain period of time and it depends on when you were born in relation to this period. I was born in the middle of the period and what they say about the Sagittarius characterizes me, however, I have acquaintances for whom these are not true"* (female). Talismans are also used by the respondents' parents, so these have a certain place in the family culture. With this type, we could not identify signs of significant inter- or intragenerational change or individual construction.

Those belonging to the self-consciously non-religious type also show signs of inter- or intragenerational stability. According to the self-consciously non-religious, respondents their rejecting standpoint towards religiosity is their choice. Religiosity is entirely absent from these families. In the early and later childhood socialization we can speak of conscious non-religion on the part of the parents as well. Rationalism and empiricism are the cornerstones for their worldview and accordingly claims that lack empirical evidence are considered false. *"I have been raised with this materialistic perspective, that the world should be explained through science and things that lack scientific results or data should be questioned, and that there is no need—I might have added this—but there is no need for me to believe in something"* (male). Their knowledge of religions is based on discussions within the family, and reading materials and is more extended than in the case of the previous two types. Experiences related to religiosity come from their community or circle of friends where they encounter youth who are considered to be religious. However, the influence of the social network does not change the self-consciously non-religious standpoint of individuals belonging to this type. Discarding religiosity is reflected on, it is an intellectual product debated with self and with others and as a result respondents highly evaluate the norm generating, integrating function of religiosity as a consequence on a social level, however, they do not desire its psychological effects on the individual level. The activities in the ritualistic dimension, the operation of churches and their representatives are the most rejected. *"That is always a common point, we agree on the ten commandments, what we don't agree on is what the source of this should be, whether the community is important, and the whole institutionalization. We don't agree on these things"* (male). In the case of the self-consciously non-religious type the family transmission of the religious standpoint is successful, recreation is not common, however, isolation from religiosity is consequent and reflected on. The intellectual curiosity of the individuals belonging to this type generates some religious knowledge and they find useful elements in the consequential dimension, however, when it comes to their worldview, experiences and rites they distance themselves from religiosity.

## 7. Conclusions

The main aim of the present research was to investigate the extent to which basic theories of religious socialization apply to young adults in the second decade of the 21st century. The database of the research is a corpus of 48 in-depth interviews. The present research focused on 18 interviews with the youngest generation in order to observe whether it is the reproduction or the constructivist theory of religious socialization that best applies to this model. Since change and stability were present simultaneously, we analyzed phenomena in the different dimensions of religiosity individually. The outcome proves that the reproduction model of religious socialization emerges, however, interestingly enough, transmission is most successful in the case of irreligiosity. When religiosity is transmitted successfully or partly successfully, reconstruction and adaptation to the individual's needs is more common. This enforces the operation of the constructivist model of religious socialization. In our analysis of complex processes, we have distinguished seven types of religious socialization based on the characteristics of the dynamics, direction and dimensions of religious socialization. Based on the outcome of religious socialization the conveniently keeping, building autonomous religiosity, the religiously impaired, drawing near, and the unreflective nostalgic types belong to the religious field, while those uninterested in the basic questions of existence and the self-consciously non-religious are characterized by irreligiosity. The reproduction model of religious socialization applies, in fact, only to the self-consciously non-religious and the uninterested. The individuals belonging to the other types reconstruct the inherited religiosity or irreligiosity under the less or more intense pressure of experiences and social networks.

The limitation of our study is that the 18 interviews do not provide enough information to get a sense of the direction of changes in the religion of a generation. Consequently, there is a need for further interviews to more accurately test the criteria put forth by our analysis, and quantitative investigations also need to be carried out. An important finding is that processes could be better described in the light of the new interpretation of religious socialization. The emergence of the active, creative, repeatedly restructuring religiosity is already a significant religious change.

The individual investigation of the dimensions of religiosity proved to be useful throughout the analysis since the dimensions are characterized by different intensity and dynamics. The ritualistic dimension was the easiest dimension to measure, while the experiential dimension was the most difficult to grasp. With some types, it was difficult to identify the experiential dimension, which contains religious feelings and the conversion experiences, mainly because these individuals were born or grew into religious life, for example the conveniently keeping, or the building autonomous religiosity types, or because the individuals did not seek the phenomenon or experienced it indirectly, for example the self-consciously non-religious and the uninterested types.

All in all, creating these types allowed us to go beyond the dichotomous and the five points religious classification. There is a need for rethinking and extending the models used so far and for creating dimensions and types which can account for current empirical experiences.

**Author Contributions:** All authors discussed the results and implications and commented on the manuscript at all stages.

**Funding:** The article was financed through the "Religious Change in Hungary" research project, supported by the National Research, Development and Innovation Office NRDI Fund, registration number K 119679.

**Conflicts of Interest:** The authors declare no conflict of interest.

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
