# Peer review of "Analysis of Religious Socialization Based on Interviews Conducted with Young Adults"

_religions, doi:10.3390/rel10060365_

Round 1
Reviewer 1 Report
MINOR REMARKS:
Under subtitle 3 (94-97) the author mention a further classification of the constructivist model into two subtypes, under subtitle 4 (110-113) the author also mention the further break down of the four dimensions on the previous the behavior patterns of the individual. In both cases, the author remains on the surface by only stating that further division is possible. It would be an added value if the author explained here why this further division creates added value for his research/article.
(127-133): The author announces one research question, but gives two. That is disturbing.
MAJOR REMARKS:
(subtitle 5, 114-133) Scarce information is provided by the author about the research methodology. According to the author the study is based on 18 interviews (out of a database of 48 interviews, which is divided into three generations). How is the database constructed (time, etc… )? Which research questions were used, based on which thematic topics/concepts? How does the author divide the pool of respondents into three generations?
(122-126) A crucial element is a result of the analysis, in particular the creation of 7 types for the interview subjects. How does the author comes to these types (by coding: open, axial, etc… )?
CONCLUSION:
In general. For a study/data-analysis based on empirical material, the author must provide additional information about 1) the research subjects, 2) the way how the database is constructed (origin, concept of generations, limitations of the database, etc… ), and 3) the process of data-analysis (coding, limitations during the data-analysis, etc… ). Without this basic information, an assessment of the research results is particularly difficult. A (minor) revision is needed for this part (in casu: subtitle 5, 114-133).
However, despite my critical reservations about the methodology, the research is particularly valuable and this explorative analysis contributes to broadening the understanding of religious socialization. As the author rightly points out, additional research will be necessary, but this is in any case an interesting start to both follow-up research and conceptual discussion.
Author Response
We are very grateful for the review provided by the reviewer of this manuscript. The comments are encouraging and the review appear to share our judgement that this study is valuable and the explorative analysis contributes to broadening the understanding of religious socialization. Please see below our responses:
Minor remarks:
(subtitle 3) The suggested corrections has been made, in both cases we have explained why the further divisions create added value.
(127-133) The correction has been made: "We created these types in order to answer the following research questions..."
Major remarks:
We have extended the 5th part, as suggested.
"The interviewees were selected by using the snowball sampling research method. As a result of this method among the interviewees were individuals from all regions of Hungary. In case of selection, the research team took into account that the individuals should belong to different groups, in terms of how determinant is their religious affiliation (the individuals are religious according to church teachings, they are religious in their own way or they are not religious at all). The semi-structured interview is also the result of joint editing. The interviews investigated three generations of the family in order to study inter- and intragenerational religious changes. The present research focuses on 18 interviews, 10 girls and 8 boys, with the youngest generation. The individuals, aged 18-30, belong to the Catholic and Reformed church".
(122-126) "One of the results of the analysis is that through the observation of the this extremely complex process we created types for the interview subjects by using content analysis, by open coding. We distinguished the following seven types: conveniently keeping, building autonomous religiosity, religiously impaired, drawing near, unreflective nostalgic, uninterested and self-consciously non-religious".
Reviewer 2 Report
This manuscript does not contain a clear introduction. It begins with "Earlier approaches in the field..." without telling us what field they are referring to. The manuscript is about religious socialization of children (the transmission of religiosity). The paper begins with a discussion of two different models of socialization: a reconstructionist and constructivist approach. The core of the paper is based on eighteen qualitative interviews. The author(s) divides them into seven types. The author(s) tell us nothing about the interviewees. They do not clearly identify to which church they belong (although in reading it, they seem to be entirely Catholic). They do not tell us in what country, what are their ages, etc. In other words, demographic information is completely lacking. How were these interviewees selected? Also, the interviews themselves are pretty superficial and do not tell us much about the process of socialization--that is how religiosity is transmitted. Based on eighteen interviews, the types also seem to be completely random with an average of less than three interviewees per type. Finally, the reference list is also very thin.
Author Response
We are very grateful for the review provided by the reviewer of this manuscript. Please see below our responses:
In the beginning we have changed the sentence, as suggested.
We have extended the research methodology with demographic information, as suggested:
"The interviewees were selected by using the snowball sampling research method. As a result of this method among the interviewees were individuals from all regions of Hungary. In case of selection, the research team took into account that the individuals should belong to different groups, in terms of how determinant is their religious affiliation (the individuals are religious according to church teachings, they are religious in their own way or they are not religious at all). The semi-structured interview is also the result of joint editing. The interviews investigated three generations of the family in order to study inter- and intragenerational religious changes. The present research focuses on 18 interviews, 10 girls and 8 boys, with the youngest generation. The individuals, aged 18-30, belong to the Catholic and Reformed church".
We have created the seven types by open coding and creating these types allowed us to go beyond the dichotomous and five points religious classification, but there is a need for further interviews and research.
Reviewer 3 Report
The intent of the study is good but I think some major clarifications need to be made. Overall, I am confused about the theories. Below I have listed the lines and my concerns.
Line 19, reference is made to the one-way linear reconstruction model and then line 20 starts with the structural functional approach. This is the only time this theory is mentioned in the paper. If the reconstruction model is based on the structural functional approach then this needs to be made more clear.
Lines 25 and 26 The constructivist and social network models are introduced. The second part of line 27 is describing these approaches but I don't know what this means. The definition of the social network model is unclear to me and it seems to be used interchangeably with the constructivist model. I think this needs clarification.
I'm unclear about what lines 32,33, and 34 are saying.
Line 38 is confusing. Reconstructing appears to be used as the newer approach.
Line 41, seven types of what? This needs to be made more clear. I would connect lines 50-52 to the previous paragraph.
Lines 55-56 need to be more clear.
Lines 66-70 there is discussion of the social network - what theory does this belong with? Not sure what line 70-71 means.
Line 79 - researcher cites our research - this needs to be a blind review.
Lines 88-90 - not sure what is being said here.
Research methods - the age of the respondents, how many are in each category. Did all of them attend denominational schools? The description of the sample needs more details. Is there a better way to identify the respondents? Male 1, Female 1 etc. This provides a better understanding of who is speaking and if people are quoted more than once.
Author Response
We are very grateful for the review provided by the reviewer of this manuscript. Please see below our responses:
Lines 25-26: The social network and the constructivist models are close to each other. Both belong to the active, correlational and open-ended model. In the case of the social network model the emphasis is on the impact of the institution.
Line 32-24: The emphasize is on the local communities, on the impact of the institution, which was missing from the previous models.
Line 41: seven types of religiosity based on our own research.
Research methods, the correction has been made, as suggested.
Round 2
Reviewer 3 Report
I have done some editing on the paper but overall I think it is good. I think it would be helpful to know how many respondents fit into each category and change the use of boy/girl to male/female. Other than a few edits and clarifications I think this paper is ready to go.

Author Response
We are very grateful for the review provided by the reviewer of this manuscript. The comments are encouraging and the review appear to share our judgement that this study is valuable and the explorative analysis contributes to broadening the understanding of religious socialization.
As the reviewer suggested we have corrected everything, we made a few edits and clarification in the manuscript.